# Exploration of the Effect of Blue Light on Functional Metabolite Accumulation in Longan Embryonic Calli via RNA Sequencing

**DOI:** 10.3390/ijms20020441

**Published:** 2019-01-21

**Authors:** Hansheng Li, Yumeng Lyu, Xiaohui Chen, Congqiao Wang, Deheng Yao, Shanshan Ni, Yuling Lin, Yukun Chen, Zihao Zhang, Zhongxiong Lai

**Affiliations:** 1Institute of Horticultural Biotechnology, Fujian Agriculture and Forestry University, Fuzhou 350002, China; lhs9897@163.com (H.L.); lyu1918@163.com (Y.L.); m18950589675_1@163.com (X.C.); w623267140@163.com (C.W.); yaodh337@163.com (D.Y.); nss951121@163.com (S.N.); buliang84@163.com (Y.L.); cyk68@163.com (Y.C.); zhangzihao863@163.com (Z.Z.); 2College of Resources and Chemical Engineering, Sanming University, Sanming 365004, China

**Keywords:** blue light, *Dimocarpus longan* Lour., embryonic callus, functional metabolites, RNA-seq

## Abstract

Light is an important factor that affects the synthesis of functional metabolites in longan embryogenic calli (ECs). However, analysis of the effect of light on functional metabolites in longan ECs via RNA sequencing has rarely been reported and their light regulation network is unclear. The contents of various functional metabolites as well as the enzymatic activities of superoxide dismutase and peroxidase and the level of H_2_O_2_ in longan ECs were significantly higher under blue light treatment than under the other treatments (dark, white). In this study, we sequenced three mRNA libraries constructed from longan ECs subjected to different treatments. A total of 4463, 1639 and 1806 genes were differentially expressed in the dark versus blue (DB), dark versus white (DW) and white versus blue (WB) combinations, respectively. According to GO and KEGG analyses, most of the differentially expressed genes (DEGs) identified were involved in transmembrane transport, taurine and hypotaurine metabolism, calcium transport and so forth. Mapman analysis revealed that more DEGs were identified in each DB combination pathway than in DW combination pathways, indicating that blue light exerts a significantly stronger regulatory effect on longan EC metabolism than the other treatments. Based on previous research and transcriptome data mining, a blue light signaling network of genes that affect longan functional metabolites was constructed and HY5, PIF4 and MYC2 were shown to be the key regulatory genes in the network. The results of this study demonstrate that the expression levels of phase-specific genes vary with changes in longan EC functional metabolites.

## 1. Introduction

Longan (*Dimocarpus longan* Lour.) belongs to *Dimocarpus*, a member of the family Sapindaceae and is a valuable fruit tree in the tropical and subtropical regions of China. Longan fruit has high nutritional and medicinal values and its main functional metabolites include polysaccharides, flavonoids, alkaloids and carotenoids [1,2]. These functional metabolites are not only physiologically important for plants but also provide medicinal, anticancer and antiaging benefits to humans [3]. Researchers have previously attempted to regulate the synthesis of functional metabolites using all types of methods; for example, adding appropriate concentrations of benzylaminopurine (BA) and kinetin (KT) to a *Taxus cuspidata* suspension cell system promoted the accumulation of paclitaxel [4], while the treatment of *Camptotheca acuminate* cells with Cu^2+^ significantly increased camptothecin production [5]. Moreover, tyrosine and tyramine have significant effects on the growth and alkaloid biosynthesis of *Lycoris radiata* suspended cells [6]. Among these methods, light is important, as it affects the accumulation of functional metabolites in plant cells. Furthermore, plant tissue and cell culture techniques are the most efficient methods for obtaining functional metabolites. Therefore, our laboratory has established an excellent longan embryogenic callus (EC) culture system to study the effect of light on functional metabolite synthesis [7].

Currently, some progress has been made in elucidating the molecular mechanisms underlying the regulatory effects of light on the production of plant functional metabolites and the mechanisms mainly include two types of control genes: structural and regulatory. Light can significantly increase the expression of structural genes in plant metabolic pathways. Azuma et al. [8] showed that light treatment significantly increases the expression of anthocyanin synthesis genes, such as chalcone synthase (*CHS*), flavonoid 3’,5’-hydroxylase (*F3’5’H*), dihydroxy flavanol reductase (*DFR*) and flavonoid 3-O-glucosyltransferase (*UFGT*) and promotes the accumulation of total anthocyanin in grape skin. Studies on bilberry (*Vaccinium myrtillus* L.) fruit have shown that light has a significant effect on the expression of carotenoid synthesis genes, such as carotenoid cleavage dioxygenase (*CCD1*) and 9-cis-epoxycarotenoid dioxygenase (*NCED1*), as well as on the total carotenoid content [9]. In addition, transcription factors (TFs) play important roles in the regulation of light and plant metabolites. Studies on litchi (*Litchi chinensis*) [10], pear (*Pyrus pyrifolia*) [11], woodland strawberry (*Fragaria vesca*) [12] and grape (*Vitis vinifera*) [13] have shown that light can induce R2R3 MYB TFs to regulate plant flavonoid metabolism. However, these mechanisms in plant functional metabolite pathways are very complex and involve a wide variety of enzymes whose synthesis is also affected by other factors. Therefore, transcriptome analysis has been applied to study the effects of light on plants.

In recent years, an increasing number of plants, such as *Arabidopsis thaliana* [14], rice (*Oryza sativa*) [14], tea plant (*Camellia sinensis*) [15], grape (*Vitis vinifera* L.) [16], potato (*Solanum tuberosum*) [17], apple (*Malus domestica*) [18] and litchi (*Litchi chinensis* Sonn.) [19], have been subjected to RNA-seq analysis to explore the effect of light on plant morphogenesis and production of functional metabolites. Light induces the transcription of many genes, inducing the differential expression of at least 20% of genes compared to dark conditions and these differentially expressed genes (DEGs) may be involved in many different physiological pathways [20,21]. In addition, some studies have revealed an initial regulatory network of light signals on plant functional metabolites. Zhang et al. [19] used transcriptome analysis to reveal the effect of light on the anthocyanin synthesis pathway of litchi pericarp. Under light conditions, the COP1/SPA (constitutive photomorphogenic 1/phytochrome A suppressor 1) complex activates photoreceptors, which rapidly passivates COP1 via direct protein-protein interactions, thereby prohibiting the degradation of the downstream bZIP TF long hypocotyl 5 (HY5) and other substrates. Furthermore, the expression of the photoreactive gene is directly controlled and the expression of the regulatory genes MYB, WD40, bHLH and others is regulated by light. These genes then bind to the structural gene promoter to jointly respond to the light signal and regulate the expression of the anthocyanin structural gene. However, studies on the transcriptome of light-to-longan EC functional metabolites have not been reported thus far and their light regulation network is unclear.

In this study, we investigated the transcriptome of longan ECs in response to different light qualities using high-throughput sequencing technology. Putative longan EC gene expression profiles were investigated under different light treatments and DEGs under different light qualities were classified. By comparing and analyzing the sequencing data of control and illuminated groups, the genes involved in the regulation of primary and secondary metabolism and their regulatory networks were established. These experiments reveal dynamic gene expression changes in response to different light qualities and provide new insights into the genetic and genomic regulation of plant functional metabolites.

## 2. Results

### 2.1. Physiological and Biochemical Indexes of Longan ECs under Different Light Qualities

Previous studies published by our laboratory showed that different light qualities affect the growth state and functional metabolites of longan ECs and revealed that blue light is optimal for the synthesis of longan EC functional metabolites [22,23]; darkness was used as a control. However, comparing blue and white light treatments is useful for understanding the difference between monochromatic and composite light and the difference in light quality. Therefore, we chose to analyze dark, blue and white light in the present study.

Light can promote the generation of polysaccharides, biotin, alkaloids and flavonoids in longan ECs and blue light promotes these four functional metabolites better than other light treatments [23]. Herein, we showed that the carotenoid content was highest (17.82 μg·g^−1^) in the blue light treatment group, followed by white light and lowest in the darkness group (11.32 μg·g^−1^) (Figure 1A and Appendix A).

The synthesis of some functional metabolites in plants is closely related to peroxidase (POD), superoxide dismutase (SOD), malondialdehyde (MDA) and H_2_O_2_ [15]. Measuring these physiological and biochemical indicators in longan ECs are helpful for understanding how the synthesis of functional metabolites is promoted by illumination. The activities of SOD and POD and the concentration of H_2_O_2_ were the highest in the blue light treatment group, followed by the white light and dark treatment groups, indicating that the light activated the longan EC enzyme antioxidant system (Figure 1B–D and Appendix A). In addition, we found no significant differences in the MDA contents under dark, blue and white light. Therefore, these results indicated that the antioxidant machinery in longan ECs had a major role in light conditions (Figure 1E and Appendix A).

In summary, three mRNA libraries (dark, blue and white) were constructed and sequenced under different light qualities to explore mRNAs related to functional metabolite biosynthesis.

### 2.2. Basic Data Analysis of Transcriptome Sequencing

To study the effects of different light qualities on functional metabolite-related mRNAs in longan ECs, three mRNA libraries were constructed and sequenced. After removing the linker sequence, the RNA-Seq data of the three longan EC libraries produced 65,116,948 to 66,451,578 reads, respectively, due to differences in light quality. In total, 86.90–87.86% of the reads were perfectly matched to the longan reference genome, while 70.30–71.55% of the reads were matched to a single locus in the longan reference genome. The sequencing results showed that these reads were well matched to the longan reference genome, providing high-quality data for further studying the longan transcriptome. Q20 and Q30 are important indicators of the quality of high-throughput sequencing, as they indicate the proportions of bases with mass fractions greater than 20% and 30%, respectively. In this study, the values of Q20 and Q30 were higher than 97% in all the longan EC samples, indicating the high reliability of the longan EC transcriptome sequencing data (Table 1).

### 2.3. Differentially Expressed Genes in Longan ECs under Different Light Qualities

To further understand the changes in the longan EC transcriptome under different light qualities, PossionD was used to calculate the expression level of each gene. The dark versus blue (DB), dark versus white (DW) and white versus blue (WB) comparisons contained 4463, 1639 and 1806 DEGs, respectively, which included 3096, 759 and 416 upregulated genes and 1367, 880 and 1390 downregulated genes (Figure 2A,C,E). Venn diagram analysis of the DB, DW and WB comparisons showed that 156 genes were commonly differentially expressed and 1819, 205 and 250 genes were unique to each comparison, respectively. Two genes were commonly upregulated, while 2497, 111 and 365 genes in each comparison were uniquely upregulated. Eighteen genes were commonly downregulated, while 636, 94 and 1317 genes in each comparison were uniquely downregulated, respectively (Figure 2B,D,F). These results indicate that the blue light significantly affected the longan ECs more than the other treatments.

### 2.4. GO Enrichment Analysis of Differentially Expressed Genes in Longan ECs

To further understand the changes in the longan EC transcriptome under light stimulation, Gene Ontology (GO) enrichment analysis was performed for three comparisons, DB, DW and WB and the results were classified into three broad categories: biological processes (BPs), cellular components (CCs) and molecular functions (MFs). The enrichment of most BPs in the DB combination was significantly higher than that in the DW combination, indicating that the effect of blue light on longan ECs was particularly obvious (Figure 3). The WB combination exhibited high enrichment, indicating that blue and white light affect longan ECs very differently (Figure 3). GO enrichment analysis indicated the top five BPs among the DB comparison DEGs were transmembrane transport, calcium ion transport, single-organism process, ion transport and phosphorylation (Figure 3 and Appendix A). In the DW comparison, the only significantly associated BP was calcium ion transduction (Figure 3 and Appendix A). The CC enrichment results in the DB combination were mainly related to the membrane, intrinsic components of the membrane, membrane parts and myosin complex (Figure 3 and Appendix A). In the DW combination, the only significantly related CC was external encapsulating structure (Figure 3 and Appendix A). GO enrichment analysis indicated that the top five MFs of DEGs in the DB comparison were ATP binding, anion binding, purine nucleoside binding, purine ribonucleotide binding and nucleoside binding (Figure 3 and Appendix A). In the DW comparison, the MFs of DEGs were mainly related to organic acid transmembrane transporter activity, oxidoreductase activity and NADPH dehydrogenase activity (Figure 3 and Appendix A). In summary, the longan ECs had substantially different responses to different light qualities and the effect of blue light on longan ECs was particularly obvious.

### 2.5. KEGG Enrichment Analysis of Differentially Expressed Genes in Longan ECs

To further understand the metabolic pathways of photoresponsive genes and other processes involved, the top 20 Kyoto Encyclopedia of Genes and Genomes (KEGG) pathways enriched among the differentially expressed mRNAs were mapped according to the enrichment factors identified (Figure 4). In the DB comparison, the top 5 enriched pathways included taurine and hypotaurine metabolism, nonhomologous end-joining, biotin metabolism, propanoate metabolism and lysine biosynthesis. In the DW comparison, the top 5 enriched pathways included taurine and hypotaurine metabolism, brassinosteroid (BR) biosynthesis, caffeine metabolism, zeatin biosynthesis and vitamin B6 metabolism. In the WB comparison, the top 5 enriched pathways included galactose metabolism, biotin metabolism, fatty acid biosynthesis, nicotinate and nicotinamide metabolism and lysine biosynthesis. The above results indicate that longan ECs responded very differently to different light qualities.

Some pathways were among the top 20 pathways enriched in all three comparisons (Table 2), such as nicotinate and nicotinamide metabolism and beta-alanine metabolism. Nicotinate and nicotinamide metabolism is closely related to alkaloid synthesis [24], while beta-alanine metabolism is closely related to biotin synthesis [25]. These results indicate that significant differences exist in the synthesis of alkaloids and biotin under dark, blue and white light conditions.

Some pathways were among the top 20 pathways enriched in two comparisons (Table 2), such as protein processing in the endoplasmic reticulum and nucleotide excision repair, indicating that these pathways may play important roles under blue and white light conditions. Galactose metabolism, fatty acid biosynthesis, lysine biosynthesis and biotin metabolism were among the top 20 pathways enriched in the DB and WB comparisons, indicating that these metabolic pathways may undergo significant changes under light conditions. Caffeine metabolism was among the top 20 pathways enriched in the DW and WB comparisons, indicating that it may change significantly under light conditions.

Some pathways were among the top 20 pathways enriched in only one comparison (Table 2). For example, taurine and hypotaurine metabolism and propanoate metabolism were among the top 20 pathways enriched in only the DB comparison, indicating that they may play specific roles in how blue light affects the synthesis of longan EC functional metabolites. Secondary metabolism, sulfur metabolism, fructose and mannose metabolism, the pentose phosphate pathway (PPP) and glycolysis/gluconeogenesis were among the top 20 pathways enriched in only the WB combination, indicating that significant differences exist in secondary metabolites, sulfur metabolism and polysaccharides in longan ECs under blue and white light conditions.

GO and KEGG enrichment analyses of the predicted mRNAs showed that light affects the synthesis of longan EC metabolites via many different mechanisms. Longan ECs might respond to different light qualities via multiple pathways, including light signal transduction, cell signal transduction, plant hormone signal transduction, photoactivation of cryptochromes (CRYs), osmotic balance, reactive oxygen species (ROS) clearance, stomatal closure and DNA repair and may play important regulatory roles.

### 2.6. Primary and Secondary Metabolic Pathways in Longan ECs under Different Light Qualities

To further understand the metabolic pathways and other processes involving longan EC DEGs, Mapman analysis was performed on the DEGs that were up and downregulated under the different light quality treatments and the metabolic pathways associated with the related genes were visually analyzed at the transcriptional level. Most of the DEGs were distributed in the cell wall, lipids, sucrose, starch and amino acid pathways as determined by Mapman analysis (Figure 5A,B). More DEGs were identified in each DB comparison pathway than in the DW comparison pathways (Figure 5A,B), indicating that blue light affected the primary metabolism in longan ECs significantly more than white light.

The fatty acid pathway is closely related to biotin synthesis [25] and the significant enrichment of lipids may affect the accumulation of biotin in longan ECs. DEGs that regulate lipids were found in both the DB and DW comparisons (Figure 5A,B) and the expression levels of most genes in the blue light treatment group were higher than those in the white light and dark treatment groups (Figure 5C). DEGs that regulate starch and sucrose were found in the DB and DW comparisons (Figure 5A,B) and the expression levels of most genes in the blue light treatment group were higher than those in the other groups (Figure 5C). These results further confirm the contribution of blue light to biotin and polysaccharide synthesis in longan ECs, which is consistent with the results of Li et al. [23].

Mapman software was used to analyze the distribution of DEGs in the secondary metabolic pathways of longan ECs, revealing significant differences. Most of the DEGs were distributed in the non-MVA, shikimate, phenylpropanoid, simple phenol, lignin and lignan, flavonoid and glucosinolate pathways (Figure 6A,B). More DEGs were identified in each DB combination pathway than in the DW combination pathways (Figure 6A,B), indicating that blue light can significantly affect the accumulation of secondary metabolism in longan ECs.

The mevalonate metabolic pathway is an upstream pathway for carotenoids and monoterpenoid alkaloids [25] and the expression levels of mevalonate metabolic pathway synthesis genes (dlo_036040.1, Dlo_027411.2, Dlo_007248.1) under blue treatment were significantly higher than those under white light and dark treatment (Figure 6C). Previous studies have demonstrated that blue light treatment does not result in a significant accumulation of total flavonoids, probably because some flavonoid pathway synthesis genes are affected by light treatment (Figure 6C). In addition, heat map analysis of DEGs demonstrated that the expression levels of most genes in secondary metabolic pathways were significantly higher under blue light treatment than under the other treatments (Figure 6C). The above results indicate that blue light is most beneficial for promoting the accumulation of carotenoids, alkaloids and total flavonoids in longan ECs, which is consistent with the results of Li et al. [23].

### 2.7. Longan EC Transcription Factors That Are Important for Photoresponsivity

In this study, a total of 1884 longan EC TFs were subjected to RNA-Seq analysis. During the longan EC photoreaction, some differentially expressed TFs appeared at a high frequency (Figure 7). For example, MYB (13.3%, v-myb avian myeloblastosis viral oncogene homologue), MYB-related (11.3%), AP2-EREBP (7.1%, APETALA21 ethylene-responsive element binding proteins), bHLH (6.3%, basic helix-loop-helix domain), NAC (6.1%, NAM, ATAF1, ATAF2 and CUC2) and MADS (6.0%, Mcm1-agrmous-deicicens-SRF4) were differentially expressed at a frequency greater than 6%, indicating that these TFs might play important regulatory roles in the synthesis of longan EC functional metabolites. TFs play important roles in photoinduced phenylpropane secondary metabolism, including MYB, MYB-related and bHLH [26]. The high frequencies of these TFs confirmed the large number of DEGs in the phenylpropane, flavonoid and lignin metabolic pathways revealed by Mapman analysis and these pathways were shown to affect the accumulation of total flavonoids in longan ECs. In addition, AP2-EREBP, bHLH and MYB can participate in hormonal signal transduction in response to the external environment [26]. In this study, most TFs, such as AP2-EREBP, bHLH and MYB, were closely related to plant hormone signal transduction. For example, phytochrome-interacting factor 4 (*PIF4*, Dlo_030081.1), an important regulatory gene for gibberellin (GA) synthesis [27], was the TF most affected by light in longan ECs. The auxin response factor (*ARF*, BGI_novel_G000726) TF, a key gene in the auxin pathway [28], was second most affected by light in longan ECs. The protein brassinosteroid insensitive 1 (*BRI1*, Dlo_031916.2) TF, important for the synthesis of BRs [29], was the 23rd most affected by light in longan ECs. These results indicate that hormonal signal transduction plays an important role in the longan EC response to light. This study also identified some bHLH TFs as important regulatory genes in the light signaling network [14]. Thus, TFs, either alone or in combination with other TFs, are involved in the effect of light on functional metabolites in longan ECs.

### 2.8. Screening and Expression Analysis of Longan EC Blue Signal-Related Genes

The above analysis confirmed the important role of light in the accumulation of metabolites in longan ECs. Based on research on *Arabidopsis* and other model plants [14,30,31,32,33,34], we explored the longan EC transcriptome data and screened out light signaling network-related genes (Figure 8). Blue light-related genes were then utilized to further verify the transcriptome sequencing results by qPCR. The expression levels of *CRY1*, *CRY2*, *COP1*, *SPA1*, *HY5*, *MYC2*, *PIF4* and *CO* were the highest in the blue light treatment group, followed by white light and dark treatment. The expression pattern of *CIB1* was the highest in white light, followed by darkness and blue light. In summary, the qPCR verification results showed that the relative expression of the genes related to the blue light network was consistent with the transcriptome sequencing results.

## 3. Discussion

### 3.1. Cell Signaling Perception and Conduction in Longan ECs under the Blue Light Condition

In this study, KEGG analysis identified the phagosome and glycosylphosphatidylinositol (GPI) anchor biosynthesis as among the top 20 most enriched pathways in only the DB comparison. GO analysis identified transmembrane transport and phosphorylation as significantly enriched in only the DB comparison. All of the above approaches involve the perception and conduction of blue light signals in longan ECs.

When plant cells are affected by external factors, the cell wall components are changed accordingly. For example, Xu et al. [35] treated *Arabidopsis thaliana* with the inhibitor isoxaben, which reduced the cell wall cellulose content, induced the expression of ET (ethylene) and JA (jasmonic acid) signaling pathway-related genes and induced the synthesis of hormones related to abiotic stress and pathogen invasion response. The external factor-sensing element is distributed at the periplasmic interval between the cell wall and the cell membrane, which may lead to an increased Ca^2+^ concentration in the cytosol [36]. The cell membrane, a barrier that prevents external substances from entering the cell freely and a carrier for exchanging information, energy and materials between cells and external factors, can maintain the stability of the intracellular environment to ensure that the physiological metabolic pathway function in the plant is executed an orderly manner [36].

GPI-anchored proteins are ubiquitously expressed on the surfaces of plant cells and their precursors are mainly synthesized by the endoplasmic reticulum and then transported to the cell membrane via the Golgi apparatus. In this process, the GPI-anchored protein is linked to the membrane by a special enzymatic reaction, sorting and selective assembly. GPI-anchored proteins are mainly associated with external factors and are involved in intracellular and cell-to-cell interactions, including signal transduction, transmembrane transport, membrane protein transport and cell attachment [37]. External factors temporarily increase the intracellular calcium concentration and induce the tyrosine phosphorylation of intracellular substrates, thereby inducing cell proliferation and differentiation [38]. The fatty acid chain linked by the GPI-anchored protein is inserted into the membrane cytoplasmic layer, interacts with molecules in the fatty acid chain (such as the signaling molecule protein tyrosine kinase (PTK)) in the cytoplasmic layer of the membrane and induces rearrangement of the outer cytoplasmic layers in the microdomain, such that the PTK (Lck, Fyn, etc.) is redistributed and polymerized in the membrane [38]. The polymerized PTK phosphorylates and activates tyrosine and tyrosine phosphorylates downstream molecules, inducing intracellular signal transduction and cell pinocytosis [38].

The longan EC cell wall is affected by blue light, resulting in dynamic changes in its composition and structure; the longan EC cell wall also maintains a uniform combination of integrity and growth. A variety of protein components are distributed at the cell wall and on the outer side of the membrane, constituting the first sensing elements that sense changes in external factors [39]. These sensors, together with polysaccharide components and receptor proteins (such as GPI-anchored proteins), initiate blue light [39]. The signal cascade is then transmitted to longan cells and the feedback mechanism regulates the expression of cell wall components and extracellular proteins [39].

### 3.2. Light Affects the Accumulation of Functional Metabolites in Longan ECs via the Ca^2+^ Signaling Pathway

GO analysis of DEGs revealed that calcium transport was significantly enriched in the DB and DW comparisons, indicating that Ca^2+^ plays a key role in longan EC photoresponsivity. The Ca^2+^ activity in plants includes nutrients and maintains the structural rigidity of the cell membrane and cell wall. In response to external factors, Ca^2+^ is also an important second messenger, as it induces the mechanisms underlying a plant’s response to the external environment [40]. The plant Ca^2+^ concentration rises instantaneously in response to external factors and can be sensed by various Ca^2+^ sensors and binding proteins, which contain aggregates of ‘EF-hand’ structural units [40]. Currently, according to their amino acid sequence similarity, tissue structure and quantity of EF-hand motifs, these proteins are divided into three categories: CaM (calmodulin)/CaM-like (calmodulin-like), CBL (calcineurin B-like protein) and CDPK (calcium-dependent protein kinase). Because CaM and CBL have no kinase activity and can transmit Ca^2+^ signals to only downstream protein kinases, they are called signal transmitters [41]. CaM can act on organelles, such as the nucleus, vacuole and cell membrane and participate in the regulation of various important physiological and biochemical processes in plants [42]. CaM, also an important regulator of plant primary and secondary metabolism, becomes active after binding to Ca^2+^ and can regulate the activities of various target proteins downstream. CaM has important biological functions, as it plays roles in cell division, enzyme activity regulation, hormone response, metabolic processes and so forth. [43]. Wang et al. [44] showed that CaM is involved in the effect of light on the synthesis of glycoalkaloids in potato tubers. Cao et al. [45] found that salicylic acid binds to plasma membrane-specific receptors, increases the intracellular calcium ion concentration and induces the activity of phenylalanine ammonia-lyase and enzyme tyrosine aminotransferase via signal cascade amplification, which allows the expression of related proteins encoded in the nucleus and ultimately promotes the synthesis of functional metabolites. Therefore, light might affect the synthesis of functional metabolites in longan ECs via calcium signaling pathways.

### 3.3. Longan ECs Initiate ROS Clearance and DNA Repair Functions in Response to Blue Light

In this study, the enzymatic activities of SOD and POD and the concentration of H_2_O_2_ in longan ECs were significantly higher under blue light treatment than under the other treatments (dark, white). KEGG analysis demonstrated that ascorbate and aldarate metabolism and nucleotide excision repair, which are related to the ROS signaling pathway and DNA repair, were among the top 20 pathways most enriched in the DB combination. ROS are normal plant cell by-products produced in some metabolic pathways or specific metabolic systems. The clearance or degradation of ROS is finely regulated and ROS are maintained at low steady-state levels. When plant cells are affected by the external environment, this metabolic balance is altered, causing excessive ROS production and damaging the plant [46]. ROS can exert a series of harmful effects on plants, as they can induce biofilm structure and functional changes, DNA strand breaks, protein denaturation and cross-linking and so forth. [47]. Plant cells mainly absorb light by DNA, which results in ROS accumulation and cellular chromosome damage, thereby also damaging genes on DNA and affecting DNA-DNA cross-linking, pyrimidine dimers and apurinic/apyrimidinic (AP) sites [47]. The first reaction to injury is repair and plants have a well-established internal protective enzyme system that counteracts the damage caused by ROS and maintains normal cellular functioning [47]. Plant cells mainly reduce oxidative damage via both enzymatic and non-enzymatic antioxidant systems; the enzymes in the enzymatic system include mainly SOD, POD, CAT and ascorbate peroxidase, while the enzymes in the non-enzymatic system include mainly low-molecular-weight antioxidants, such as carotenoids, flavonoids, ascorbate and glutathione [48]. Efficient removal of ROS from plants requires that various antioxidants and antioxidant enzymes act synergistically. In the cytoplasm, chloroplasts, mitochondria and peroxisomes, SOD can result in disproportionate O^2−^ accumulation, leading to the production of antioxidant enzymes that maintain high activity. These phenomena improve the oxidative damage repair capacity and ROS scavenging ability of the plant cell and thereby enhance the plant’s ability to cope with external factors [49].

We herein demonstrated that blue light significantly increased the activities of POD and SOD and the concentration of H_2_O_2_ as well as the polysaccharide, biotin, carotenoid, alkaloid and total flavonoid contents in longan cells. These results indicate that under blue light conditions, H_2_O_2_ burst occurs, cells undergo oxidative stress and defensive reactions occur via intracellular signaling, leading to the cellular accumulation of some functional metabolites. In response to blue light, the activities of POD and SOD in longan cells are increased to prevent oxidative damage and thus cope with the induced intracellular physiological changes. At the same time, the DNA repair pathway is activated during the blue light response.

### 3.4. The Important Role of the Sulfur Metabolism Pathway in Longan EC Light-Affecting Functional Metabolites

KEGG analysis demonstrated that taurine and hypotaurine metabolism were among the foremost pathways enriched in the DB and DW comparison, while sulfur metabolism was among the top 20 pathways enriched in the WB comparison. Sulfinous sulfonic acid is an S^2−^ scavenger and the sulfinic acid content in plants affects the entire sulfur metabolism pathway [50]. As one of the basic elements of plant growth and development, sulfur is a constituent of amino acids and proteins, a key medium for the synthesis of chlorophyll, coenzyme and glutathione and an essential element for enzymatic activity. Sulfur also participates in plant BPs, as it plays roles in cytoplasmic membrane structure and functional expression, growth regulation, respiration, photosynthesis, stress resistance and metabolism [51,52]. The main forms of sulfur in plants include organic sulfur compounds and inorganic sulfates (SO_4_^2−^). Organic sulfur compounds, such as cystine and cysteine, are mostly found in plant organs and inorganic sulfates mostly accumulate in the vacuoles of cells in the form of SO_4_^2−^ [50].

External environmental factors result in the plant production of sulfur derivatives (reactive sulfur species, RSS) and ROS, which damage cell membranes, proteins and DNA, leading to metabolic disorders and the activation of defense mechanisms. As the primary redox buffer, glutathione plays an important role in a plant’s response to the external environment [50]. In plant cells, H_2_S is produced by the breakdown of cysteine, an intermediate cellular product that serves as the hub of the entire sulfur element transformation pathway. The decomposition of cysteine produces H_2_S, accompanied by the production of pyruvic acid, an important substance in the glycolysis pathway during carbon metabolism that participates in the conversion of carbon [53,54]. Sun et al. [55] found that H_2_S can regulate carbon and sulfur metabolism in corn and promote the accumulation of triterpenoids.

In the response of longan ECs to light, the expression levels of cysteamine dioxygenase and glutamate decarboxylase were downregulated in the taurine and hypotaurine metabolism pathways, resulting in reduced taurine and hypotaurine (S^2−^ scavenger) content and increased S^2−^ content. 3’-Phosphoadenosine 5’-phosphosulfate synthase and adenylyl-sulfate reductase were significantly upregulated in the sulfur metabolic pathway, resulting in increased adenylyl-sulfate and S^2−^. Increased S^2−^ concentrations might affect the biotin, fatty acid, mevalonate and steroid synthesis pathways in longan ECs [55].

### 3.5. Longan ECs Respond to Light via the TCA and PPP Pathways

Nicotinate and nicotinamide metabolism; lysine biosynthesis; valine, leucine and isoleucine degradation; propanoate metabolism; and galactose metabolism were among the top 20 pathways enriched in the DB comparison as determined by KEGG analysis. KEGG analysis demonstrated that pyruvate metabolism and alanine, aspartate and glutamate metabolism were among the top 20 pathways enriched in the DW comparison. These metabolic pathways involve mainly the tricarboxylic acid cycle (TCA) and pentose phosphate pathway (PPP). Carbohydrate catabolism is the most susceptible to the external environment and involves three pathways: the PPP, the TCA cycle and the Emben-Meyerhof-Parnas (EMP) pathway. The glycolytic pathway is responsible for the conversion of glucose to H+ and pyruvate, accompanied by the release of high-energy substances (ATP and NADH), constituting the first stage of plant cell respiration [56]. The TCA cycle, the second stage of cellular respiration, mainly degrades organic matter and adsorbs energy in the form of oxygen to meet the needs of cell growth and division [57,58]. The PPP, located in the bypass of the TCA cycle, produces pentose and NADPH, thus providing matrix material and reducing power for the biosynthesis of plant cell materials. The main function of the PPP is to participate in anabolism rather than catabolism [59,60]. The significant enrichment of the PPP and TCA cycle in longan ECs promotes the synthesis of a large amount of acetyl-CoA, which might lead to increased contents of carotenoids, monoterpenoid alkaloids, steroids and so forth. via the mevalonate/2-C-methyl-D-erythritol 4-phosphate (MVA/MEP) pathway and these substances enhance ROS clearance in response to light.

### 3.6. Some Metabolic Pathways are Closely Related to the Synthesis of Biotin, Polysaccharides, Carotenoids, Alkaloids and Total Flavonoids in Longan ECs

In the response of longan ECs to light, the galactose, fructose and mannose metabolism, PPP, glycolysis/gluconeogenesis and N-glycan biosynthesis pathways as well as other types of O-glycan biosynthesis pathways were among the top 20 enriched pathways and all of these pathways are closely related to the synthesis of sugar metabolism. These results indicate that light promotes the synthesis of polysaccharides in longan ECs (Figure 9). The β-alanine, propanoate, pyruvate, fatty acid, unsaturated fatty acid and biotin metabolic pathways, which all significantly affect the synthesis of acetyl-coenzyme A and malonyl coenzyme A, were among the top 20 enriched pathways. Acetyl-coenzyme A is a pivotal product in the metabolism of energy substances in plants and the three major nutrients proteins, fats and sugars are aggregated into a common metabolic pathway [61]. Acetyl-coenzyme A is a precursor of synthetic ketone bodies, fatty acids, monoterpenoid alkaloids, steroids, carotenoids and so forth. [25]. For example, acetyl-CoA C-acetyltransferase, malonate-semialdehyde dehydrogenase and acetyl-CoA carboxylase function in propanoate metabolism; malealdehyde-semialdehyde dehydrogenase functions in β-alanine metabolism; and acetyl-CoA carboxylase functions in fatty acid biosynthesis. The abovementioned longan EC DEGs were significantly upregulated during the response to light, which increased the accumulation of acetyl-coenzyme A. As a precursor of the two pathways, acetyl-coenzyme A promotes the synthesis of malonyl coenzyme A, which affects the fatty acid and ketone pathways, thereby promoting the accumulation of biotin; on the other hand, acetyl-coenzyme A affects the mevalonate pathway and promotes the synthesis of carotenoids and monoterpenoid alkaloids (Figure 9) [25]. Nicotinate and nicotinamide metabolism, lysine metabolism and other pathways were among the top 20 enriched pathways. Niacin and lysine are precursors of alkaloid synthesis [24], indicating that light promotes the synthesis of alkaloids in longan ECs and that their synthesis can affect the accumulation of flavonoids (Figure 9). Therefore, the above conclusions might explain the accumulation of biotin, polysaccharides, carotenoids, alkaloids and total flavonoids in longan ECs under light conditions.

### 3.7. Blue light Signaling Network in Longan Functional Metabolites

Light plays an important role in plant primary and secondary metabolic synthesis pathways. In a study on Arabidopsis, the blue light signaling network, which includes the COP1/SPA, CIB1 and phytochrome-interacting factor (PIF) pathways, was determined to affect light morphogenesis (Figure 10) [14,30,31,32,33,34].

COP1/SPA pathway: CRYs also interact with the SPA1/COP1 complex at the post-transcriptional level to indirectly regulate gene expression. Arabidopsis CRY1 and CRY2 interact with SPA1 only in response to blue light and not in response to red light and CRY1 inhibits the degradation of the HY5, HYH (HY5 homologue) and HFR1 (long hypocotyl in far-red 1) proteins under blue light [31,34]. These TFs regulate the expression of removed etiolation genes and CRY2 also inhibits the degradation of the TF CO protein by COP1 under blue light [14,34]. CRYs inhibit blue light-dependent COP1 activity and promote CO accumulation and flower development initiation to conduct photoperiodic signals [62]. That is, the interaction of CRYs with the SPA protein inhibits the SPA activation of COP1 activity, thereby inhibiting the degradation of HY5, HYH, CO and other transcriptional regulators by COP1 and promoting the expression of light-regulated genes.

CIB1 pathway: CIB1 was the first blue light-dependent CRY2-interacting protein found in plants [63]. *Arabidopsis thaliana* CRY2 interacts with the bHLH TF CIB1 under blue light and transcription is regulated by interaction with the TF CIB1 and other CIBs.

PIF pathway: Ma et al. [64] and Pedmale et al. [65] demonstrated that CRY1 and CRY2 bind to PIF4 and PIF5 under light conditions and CRY1 can also bind to PIF3. The PHR domain of CRY2 binds to half of the PIF5 N-terminus, an independent APB motif, indicating that CRY2 and phyB recognize different structural PIFs [65]. The interaction between CRY and PIFs regulates the low blue light (LBL)-induced shading response (SAR) and temperature-induced cell growth [64,65]. Unlike the CRY2-CIB interaction, the CRYs-PIF4/5 interaction inhibits PIF4/5 activity [64,65].

This study found blue light to be the most beneficial among all the lights studied for promoting the accumulation of polysaccharides, biotin, carotenoids, alkaloids and total flavonoids in longan ECs. Therefore, based on the research on Arabidopsis and other model plants, we searched the longan transcriptome, screened out light signaling network-related genes (Figure 8) and constructed an initial blue light signaling network of genes that affect longan functional metabolites (Figure 10).

Plants can accurately sense light conditions ranging from UV-B to far-red light via a variety of photoreceptors to coordinate the response to light and their response to blue light is mediated mainly via CRY photoreceptors [66]. Most plants contain several CRY genes and *CRY1*, *CRY2* and *CRY-DASH* have been isolated and identified in Arabidopsis [67]. *CRY1* and *CRY2* have also been confirmed in longan. We herein showed that blue and white light can promote the expression of *CRY1* (Dlo_007957.1) and *CRY2* (Dlo_022523.1) and the expression of these genes in response to blue light was higher than that in response to other treatments (Figure 8Ba,b).

COP1 is a photomorphogenesis inhibitor that plays a decisive role in light signal transduction [68]. The molecular switch located downstream of CRYs plays an important role in light-induced plant photomorphogenesis processes, including flowering [69], growth and development [70] and primary and secondary metabolism [71]. Under light conditions, COP1 and SPA can form complexes in the nucleus and regulate the synthesis of plant functional metabolites [72]. This study demonstrated that the expression of *COP1* (Dlo_ 017664.1) was significantly upregulated under light conditions and its expression under blue light treatment was slightly higher than that under white light treatment (Figure 8Bc), which was consistent with the trend of polysaccharide, biotin, alkaloid and total flavonoid contents in longan ECs.

*HY5*, a key factor in light signal transduction, also plays an important role in the regulation of plant functional metabolites. Research on Arabidopsis has confirmed that blue light can regulate the expression of *PAP1* (an R2R3-MYB TF) via the light signal TF *HY5* and then regulate flavonoid synthesis [73]. The accumulation of anthocyanin in apple is significantly correlated with the expression of *HY5* [74]. In this study, the expression level of *HY5* (Dlo_017904.1) was consistent with the trend of functional metabolite contents in longan ECs (Figure 8Be).

*MYC2* (bHLH TF) is a node that regulates plant metabolic synthesis by blue light signals [75]. In a study on *Catharanthus roseus*, *MYC2* was found to promote the accumulation of alkaloids [75,76] and *Arabidopsis thaliana MYC2* was shown to positively regulate the biosynthesis of flavonoids by positively regulating other TFs. In contrast, *MYC2* negatively regulates the biosynthesis of the JA-responsive tryptophan derivative steroidal glucosinolate [77]. In this study, *MYC2* (Dlo_012527.1) was significantly upregulated under blue and white light treatment (Figure 8Bf). Therefore, *MYC2* might act as a positive regulator to promote the synthesis of longan polysaccharides, biotin, alkaloids and total flavonoids under light conditions, especially in blue light.

PIFs play a central role in the photoregulation of plant growth and metabolism [78]. Arabidopsis studies have shown that *PIF1* and *PIF3* can negatively regulate the synthesis of chlorophyll [79,80], while *PIF3* positively regulates the accumulation of anthocyanins [81]. A transcriptome analysis of light quality on the Norwegian spruce (*Picea abies* (L.) Karst.) revealed that blue light promotes the upregulation of *PIF3*, which, in turn, regulates the synthesis of flavonoid metabolism [82]. *PIF4* was significantly upregulated in blue and white light and *PIF4* was confirmed to be a positive regulator of longan metabolite synthesis (Figure 8Bg).

In summary, the results of this study lead to the speculation that blue light can affect the synthesis of functional metabolites in longan via three pathways (Figure 10). First, under the action of blue light, the interaction of CRYs with the SPA protein inhibits the ability of SPA to activate COP1, inhibiting the degradation of HY5 and other TFs by COP1 and upregulating light-regulated genes, thereby promoting the accumulation of longan functional metabolites. Alternatively, COP1 can also directly act on the MYB-bHLH-WD40 TF to regulate metabolic pathways. Second, functional metabolite synthesis is regulated by the TF *MYC2*. Third, CRYs bind to PIF4 and other members of the PIF family to regulate functional metabolite synthesis.

## 4. Materials and Methods

### 4.1. Plant Material and Light Treatments

Longan ECs were obtained using the method provided by Lai [7]. Based on previous studies on longan ECs, Murashige and Skoog (MS) medium supplemented with 20 g·L^−1^ sucrose, 6 g·L^−1^ agar and 4.5 µM 2,4-dichlorophenoxyacetic acid (2,4-D) (pH 5.8) was utilized for the light treatment experiments. Longan ECs were incubated in a plant light chamber and control groups were placed in the dark. The intensity and photoperiod of blue (457 nm) and white light were fixed at 32 µmol·m^−2^·s^−1^ and 12 h·Day^−1^, respectively. Equal pieces (0.04 g) of longan ECs (5–7 mm in diameter) were placed in separate bottles (240 mL); thirty bottles were included per treatment condition. Treatment was conducted for 25 d at 25 ± 2 °C with a relative humidity of 55–60%, after which longan ECs were stored at −80 °C for later use.

### 4.2. RNA-Seq Library Construction

Total RNA was isolated from longan ECs with the TriPure Isolation Reagent (Roche Diagnostics, Indianapolis, IN, USA) according to the manufacturer’s instructions. The integrity and concentration of the RNA samples were further measured using an Agilent 2100 Bioanalyzer (Agilent, CA, USA) and the purity of the RNA samples was assessed using the NanoPhotometer^®^ spectrophotometer (NP80, IMPLEN, Munich, DE). RNA libraries were prepared using the True-seq RNA sample preparation kit according to the manufacturer’s instructions. The constructed library was tested on the Agilent 2100 Bioanalyzer and the ABI StepOnePlus Real-Time PCR System. Finally, small RNA libraries were sequenced on an Illumina HiSeq 4000 platform (Shenzhen, China).

### 4.3. Mapping Reads to the Reference Genome

The clean reads were mapped to the longan reference genome for statistical analysis and sequences were aligned using Hierarchical Indexing for Spliced Alignment of Transcripts (HISAT) software [83].

### 4.4. Quantification of Gene Expression Levels

Clean reads were mapped to the longan reference genome using Bowtie v2.2.3 software [84]. The expression levels of genes and transcripts were calculated using RNA-Seq with RSEM v1.2.12 software [85].

### 4.5. Differential Expression Analysis

DEGs were identified using the PossionDis method [86]. Briefly, read counts were estimated based on the fragments per kilobase million (FPKM) method using Jakhesara’s criteria to calculate the expression level of each gene in longan ECs. Differential expression analysis of the three treatments was performed using the DEGSeq R package. P values were adjusted using the Benjamini and Hochberg method. A corrected P-value (false discovery rate, FDR) of 0.001 and a fold change of 2 were set as the default threshold for defining significant differential expression.

### 4.6. GO and KEGG Enrichment Analyses of Differentially Expressed Genes

According to the GO and KEGG annotation results and the official classification, DEGs were classified and the enrichment factors were analyzed using the phyper function in R software.

### 4.7. Carotenoid Determination

Carotenoid contents were measured using the method provided by Shao [87] with some modifications. Briefly, longan ECs were freeze-dried for two days in a refrigerant dryer (LGJ-25C, Sihuan, Beijing, China) and 0.2 g of fine longan EC powder was dissolved in 10 mL of acetone. Then, longan ECs were completely converted to a white color under dark conditions. The supernatant was collected into a new tube and fixed with 10 mL of acetone. Finally, the absorbance of the extract was measured at a wavelength of 450 nm using an ultraviolet-visible spectrophotometer (T6, Puxi, Beijing, China). The carotenoid contents in longan ECs were calculated according to Shao’s method.

### 4.8. Measurement of Antioxidant Enzymes

Briefly, 0.1 g of longan ECs (fresh weight) from each of the treatment groups was ground in liquid nitrogen. After 1 mL of the extract was added, the homogenate was centrifuged at 8000× *g* for 10 min at 4 °C and the supernatant was removed and placed on ice for testing. Finally, the supernatant was collected for enzyme activity assays. SOD, POD, MDA and H_2_O_2_ were assayed using commercial kits (Keming, Suzhou, China) and an ultraviolet-visible spectrophotometer according to the manufacturer’s instructions and a previous report.

### 4.9. Validation of the DEGs by qRT-PCR

To validate the RNA-Seq results, 9 DEGs were subjected to qRT-PCR analysis performed on a LightCycler480 real-time PCR system (Roche, Basel, Switzerland). Relative gene expression levels were evaluated according to a previous method [88]. The changes in mRNA expression were calculated using the comparative 2^−ΔΔ^Ct method. Specific primers were designed using DNAMAN V6.0; the primer pair sequences are listed in Appendix A. All treatments were analyzed in three biological replicates.

### 4.10. Statistical Analysis

Quantitative results for physiological and biochemical indexes and gene expression analyses are presented as the means ± SDs of at least three biological replicates. The effects of different light qualities on physiological and biochemical indexes and gene expression of longan ECs were analyzed by one-way analysis of variance (ANOVA) followed by Duncan’s test using SPSS version 19.0. Figures were prepared using Omicshare online software and GraphPad Prism 6.0 software.

## Figures and Tables

**Figure 1 ijms-20-00441-f001:**
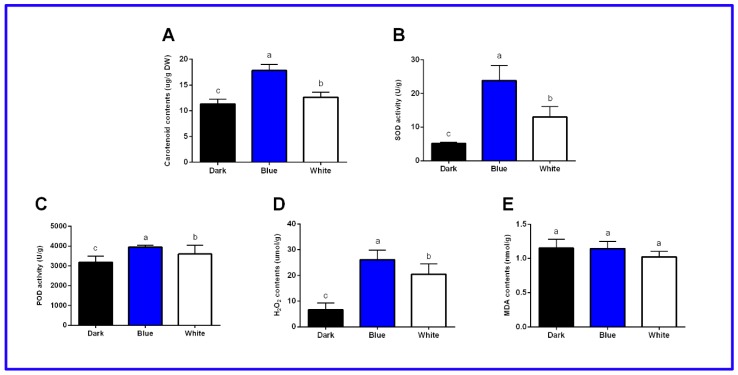
Physiological and biochemical indicators of longan ECs under different light qualities. **A**, Changes in carotenoid contents under different light qualities. **B**, SOD activities in longan ECs under different light qualities. **C**, POD activities in longan ECs under different light qualities. **D**, H_2_O_2_ contents in longan ECs under different light qualities. **E**, MDA contents in longan ECs under different light qualities. Values represent means ± SDs of three replicates. Different lower-case letters indicate statistically significant differences at the 0.05 level by one-way ANOVA with Duncan’s test.

**Figure 2 ijms-20-00441-f002:**
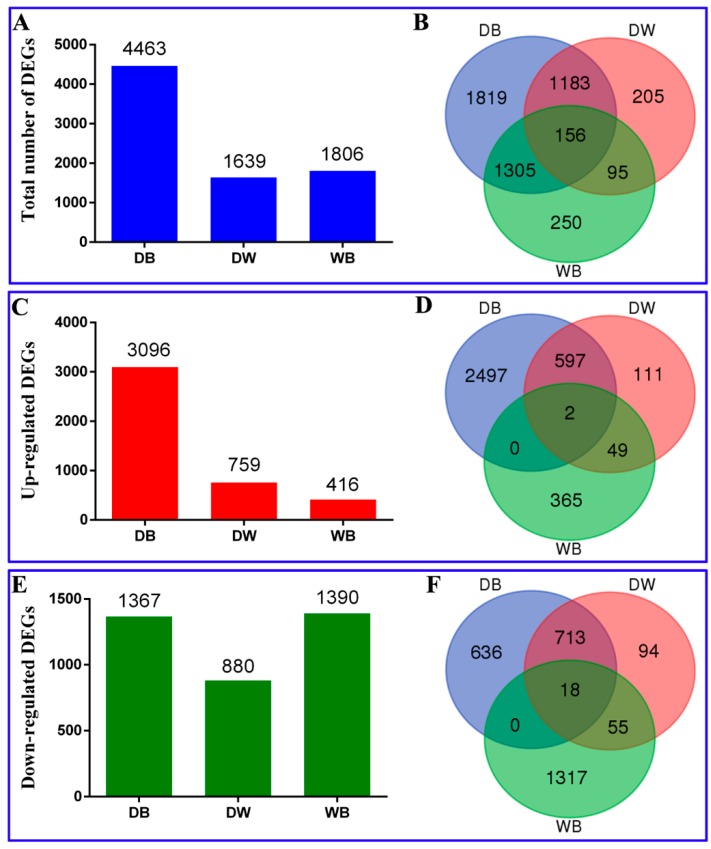
Differentially expressed mRNAs in longan ECs. **A**, The number of differentially expressed mRNAs in response to different light qualities (fold change ≥ 2.00 and FDR ≤ 0.001). **B**, Venn diagram representing the uniquely and commonly regulated mRNAs in longan ECs under different light qualities. **C**, Number of mRNAs up-regulated in response to different light qualities. **D**, Venn diagram representing the uniquely and commonly regulated mRNAs up-regulated in the longan ECs under different light qualities. **E**, Number of mRNAs down-regulated in response to different light qualities. **F**, Venn diagram representing the uniquely and commonly regulated mRNAs down-regulated in the longan ECs under different light qualities; The control samples treated with darkness were called ‘D.’ ‘B’ indicates blue light treatment and ‘W’ indicates white light treatment. ‘DB’ indicates dark versus blue, ‘DW’ indicates dark versus white and ‘WB’ indicates white versus blue.

**Figure 3 ijms-20-00441-f003:**
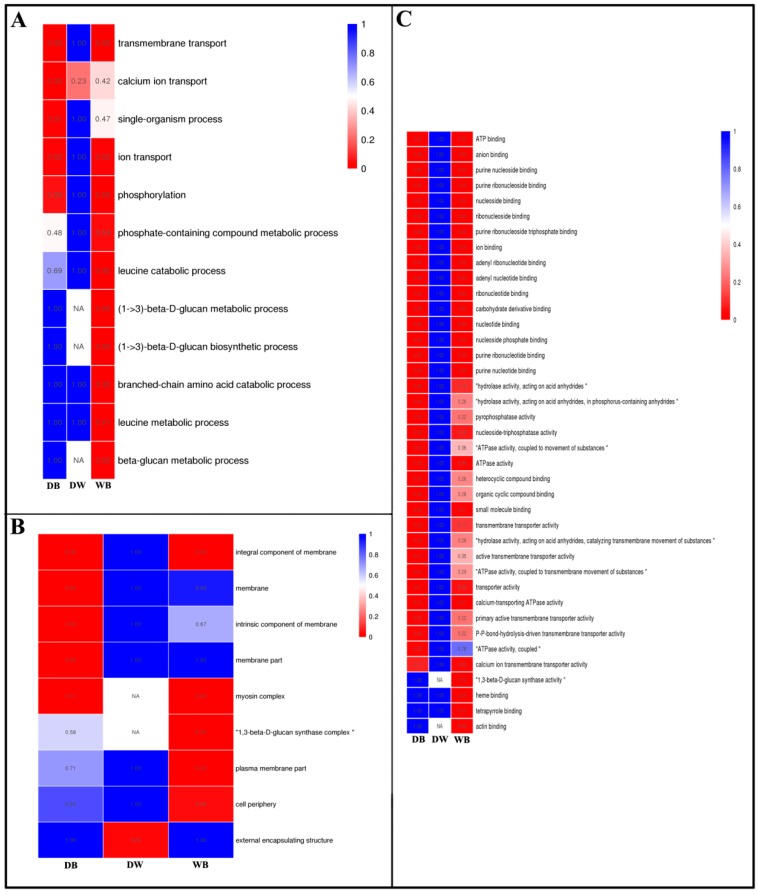
GO term enrichment analysis of differentially expressed genes in longan ECs under different light qualities. From the red to the blue corresponds to the numerical value of corrected P and significant enriched GO terms from the low to the high. **A**, biological process; **B**, cellular component; **C**, molecular function.

**Figure 4 ijms-20-00441-f004:**
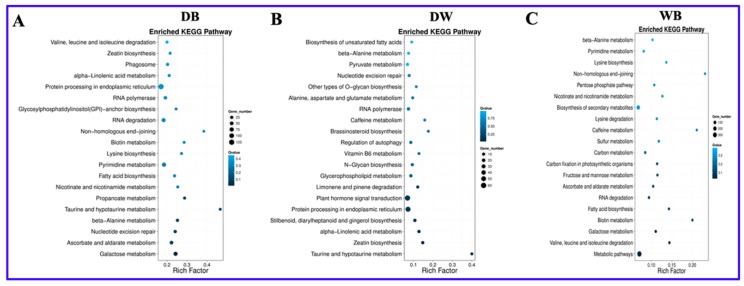
KEGG pathway analysis was employed in A, B, and C to fully characterize these differentially expressed genes. A, DB; B, DW; C, WB; The Y-axis on the left represents KEGG pathways and the X-axis indicates the richness factor. Low q-values are shown in dark blue and high q-values are depicted in light blue.

**Figure 5 ijms-20-00441-f005:**
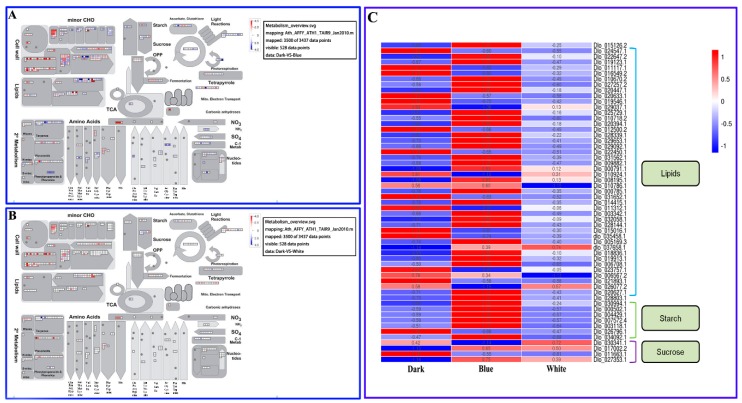
Longan EC metabolic pathways under different lighting conditions. **A**, Metabolic pathways of differential target genes in DB. **B**, Metabolic pathways of differential target genes in DW. **C**, Heat maps of expression of differentially expressed target genes under different lighting conditions. From red to blue corresponds to a decreasing value of FPKM.

**Figure 6 ijms-20-00441-f006:**
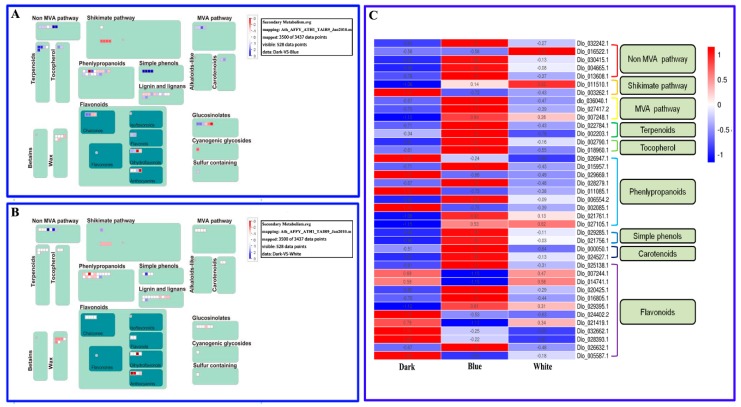
Longan EC secondary metabolic pathways under different lighting conditions. **A**, Secondary metabolic pathways of differential target genes in DB. **B**, Secondary metabolic pathways of differential target genes in DW. **C**, Heat maps of expression of differential target genes under different lighting conditions.

**Figure 7 ijms-20-00441-f007:**
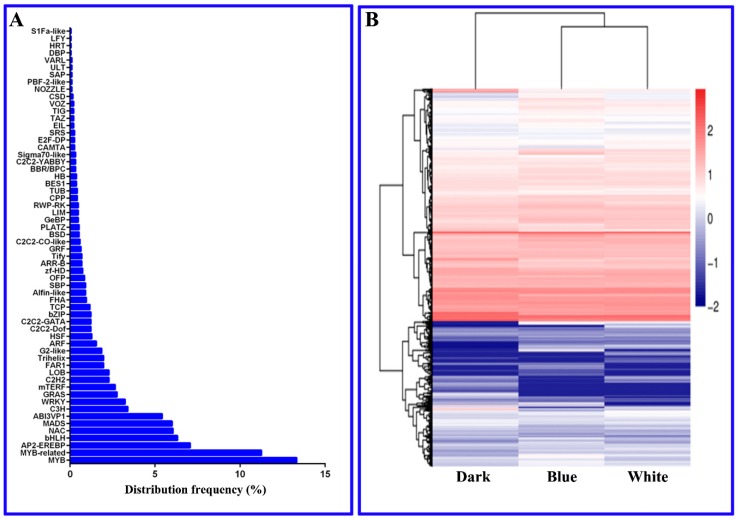
**A**, Distribution of TF frequencies; **B**, Heat map of TF expression patterns. The shift from red to blue corresponds to a decreasing value of FPKM.

**Figure 8 ijms-20-00441-f008:**
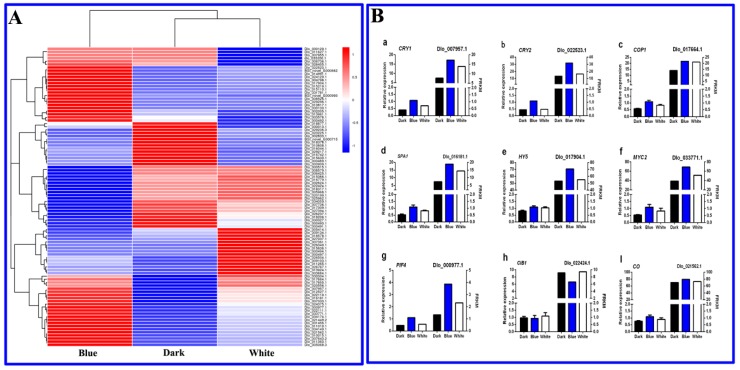
(**A**) Heat maps of the blue light signaling networks in up- and downregulated genes. (**B**) qRT-PCR analysis of candidate genes involved in the blue light stress response. From red to blue corresponds to a decreasing value of FPKM. CRY1, cryptochrome 1; CRY2, cryptochrome; COP1, constitutive photomorphogenic 1; HY5, long hypocotyl 5; MYC2, basic helix-loop-helix transcription factor; PIF, phytochrome-interacting factor; SPA1, phytochrome A suppressor 1; CIB1, cryptochrome-interacting basic-helix-loop-helix 1; CO, constants.

**Figure 9 ijms-20-00441-f009:**
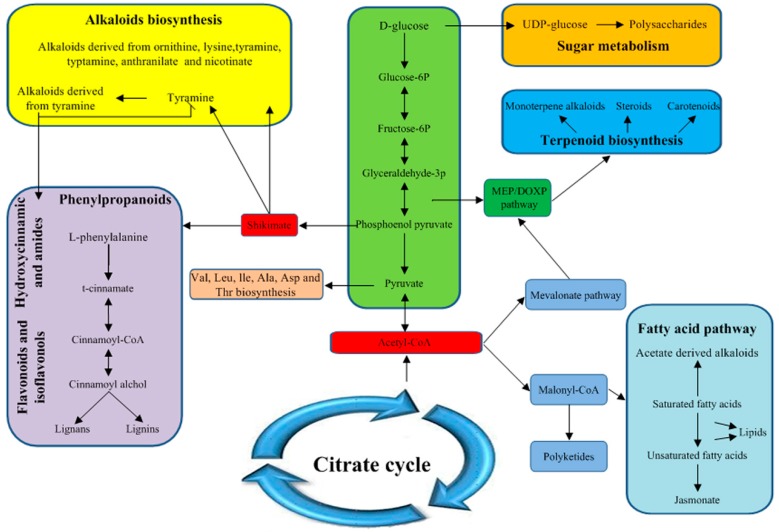
Satellite metabolic pathways involved in the biosynthesis of light-related metabolites in longan ECs.

**Figure 10 ijms-20-00441-f010:**
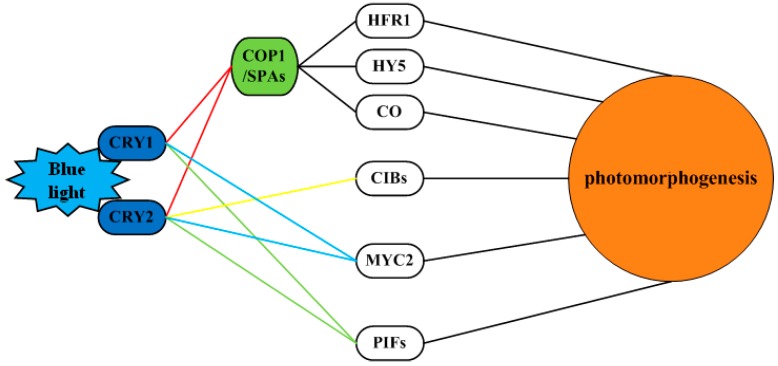
Blue light signaling network of genes underlying photomorphogenesis in longan.

**Table 1 ijms-20-00441-t001:** Summary of the sequencing data in each sample.

Sample	Dark	Blue	White
Total Raw Reads (Mb)	74.35	74.35	74.35
Total Clean Reads	66,451,578	65,308,954	65,116,948
Total Clean Bases (Gb)	6.65	6.53	6.51
Q20 (%)	99.20	99.26	99.28
Q30 (%)	97.26	97.43	97.47
Total Mapping Ratio (%)	86.90	87.41	87.86
Uniquely Mapping Ratio (%)	70.30	71.21	71.55

**Table 2 ijms-20-00441-t002:** Top 20 KEGG pathways enriched by DEGs in the 3 groups.

	Pathway Term	Group		
1	Nicotinate and nicotinamide metabolism	DB	DW	WB
2	β-Alanine metabolism	DB	DW	WB
3	RNA polymerase	DB	DW	
4	Protein processing in endoplasmic reticulum	DB	DW	
5	α-Linolenic acid metabolism	DB	DW	
6	Nucleotide excision repair	DB	DW	
7	Zeatin biosynthesis	DB	DW	
8	Galactose metabolism	DB		WB
9	Ascorbate and aldarate metabolism	DB		WB
10	Fatty acid biosynthesis	DB		WB
11	Pyrimidine metabolism	DB		WB
12	Lysine biosynthesis	DB		WB
13	Biotin metabolism	DB		WB
14	Nonhomologous end-joining	DB		WB
15	RNA degradation	DB		WB
16	Valine, leucine and isoleucine degradation	DB		WB
17	Caffeine metabolism		DW	WB
18	Taurine and hypotaurine metabolism	DB	DW	
19	Propanoate metabolism	DB		
20	Glycosylphosphatidylinositol (GPI)-anchor biosynthesis	DB		
21	Phagosome	DB		
22	Stilbenoid, diarylheptanoid and gingerol biosynthesis		DW	
23	Plant hormone signal transduction		DW	
24	Limonene and pinene degradation		DW	
25	Glycerophospholipid metabolism		DW	
26	N-Glycan biosynthesis		DW	
27	Vitamin B6 metabolism		DW	
28	Regulation of autophagy		DW	
29	Brassinosteroid biosynthesis		DW	
30	Alanine, aspartate and glutamate metabolism		DW	
31	Other types of O-glycan biosynthesis		DW	
32	Pyruvate metabolism		DW	
33	Biosynthesis of unsaturated fatty acids		DW	
34	Metabolic pathways			WB
35	Biosynthesis of secondary metabolites			WB
36	Fructose and mannose metabolism			WB
37	Carbon fixation in photosynthetic organisms			WB
38	Sulfur metabolism			WB
39	Pentose phosphate pathway			WB
40	Glycolysis/Gluconeogenesis			WB

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
