# Peer review of "Exploration of the Effect of Blue Light on Functional Metabolite Accumulation in Longan Embryonic Calli via RNA Sequencing"

_ijms, 2019, doi:10.3390/ijms20020441_

Round 1
Reviewer 1 Report
Hansheng Li, et al in this manuscript explore the transcriptional changes mediated by using dark white and blu lights on growing coltures of longan EC.
The work is well presented and might be suitable for publication after some minor changes.
1) The title "Exploration of the effect of blue light on longan embryonic callus functional metabolite accumulation via RNA sequencing " is misleading since in this manuscript only the carotenoids have been quantified and this result is indeed available in the supplementary material. Albeit a similar title has been used for other two papers from the same authors, in reviewer's opinion the analysis of carotenoids cannot be defined as an exploration on functional metabolite accumulation.
2) Similarly as above, in the text in several points the authors declare that results indicate blu light as beneficial for functional metabolites accumulation (just few examples i.e line 263-264, 410-412, 459-462). In this work these conclusions appear speculative not being supported by quantitative analysis of alkaloids , terpenoids and so on , therefore please rearrange the text according to the results here presented or refer to sperimental results already published, if so .
3) In this manuscript is not reported which is the actual metabolite production of longan EC or the metabolite production under different lights, this cannot be deduced by the up or -downregulation of metabolic pathways or gene expression , the reviewer suggests to add a supplementary table with the major metabolites produced by longan EC in control condition , it would be a great improvment if a metabolic comparison of the longan Ec under different lights could be reported .
4) Antioxidant enzymes are high in EC under blu light , this finding is interesting , but we do not know whether stress symptoms occurred or the antioxidant machinery is enought efficient to counteract the stress , don't you think that to perform a MDA assay or a ROS detection, might be a step forward?
Author Response
Responses to the reviewers’ comments
The reviewer’s questions have been provided below, followed by our point-by-point responses. All of the modifications are highlighted in yellow in the revised manuscript. We appreciate the reviewers’ and editor’s insightful comments and are grateful for the time they have spent reviewing our manuscript.
Reviewer #1
1) The title "Exploration of the effect of blue light on longan embryonic callus functional metabolite accumulation via RNA sequencing" is misleading since in this manuscript only the carotenoids have been quantified and this result is indeed available in the supplementary material. Albeit a similar title has been used for other two papers from the same authors, in reviewer's opinion the analysis of carotenoids cannot be defined as an exploration on functional metabolite accumulation.
Reply: Lines 97-98 of this manuscript cite reference [23], whose first author is myself. That paper published polysaccharide, biotin, alkaloid, and flavonoid contents of longan ECs in control, blue and white light. The results have shown that light can promote polysaccharides, biotin, alkaloids and flavonoids in longan ECs, and blue light promotes these four functional metabolites better than other treatments (Figure 1). Therefore, we have completed the determination of polysaccharides, biotin, alkaloids, flavonoids and carotenoids in longan ECs.
[23] Li, H.; Chen, X.; Wang, Y.; Yao, D.; Lin, Y.; Lai, Z., Exploration of the effect of blue light on microRNAs involved in the accumulation of functional metabolites of longan embryonic calli through RNA-sequencing. Journal of the Science of Food & Agriculture 2018.
Figure 1. Metabolic contents of longan ECs under different lighting conditions. Global analysis of sRNA libraries from longan ECs. (A) Changes in polysaccharide contents under different light treatments. (B) Changes in biotin contents under different light treatments. (C) Changes in alkaloid contents under different light treatments. (D) Changes in flavonoid contents under different light treatments. Values represent the means ± SDs of three replicates. Different upper-/lowercase letters indicate statistically significant differences at the 0.01/0.05 levels by one-way ANOVA and Duncan’s test.
2) Similarly, as above, in the text in several points the authors declare that results indicate blue light as beneficial for functional metabolites accumulation (just few examples i.e line 263-264, 410-412, 459-462). In this work, these conclusions appear speculative not being supported by quantitative analysis of alkaloids, terpenoids and so on, therefore please rearrange the text according to the results here presented or refer to sperimental results already published, if so.Reply: We have rearranged the text and cited reference [23].
These results further confirm the contribution of blue light to biotin and polysaccharide synthesis in longan ECs, which was consistent with the results of Li et al [23]. (manuscript lines 245-247)
The results also indicate that blue light was most beneficial for promoting the accumulation of carotenoids, alkaloids and total flavonoids in longan ECs, which was consistent with the results of Li et al [23]. (manuscript lines 267-269)
The manuscript cited reference [23], which has published polysaccharide, biotin, alkaloid, and flavonoid contents of longan ECs in control, blue and white light.
[23] Li, H.; Chen, X.; Wang, Y.; Yao, D.; Lin, Y.; Lai, Z., Exploration of the effect of blue light on microRNAs involved in the accumulation of functional metabolites of longan embryonic calli through RNA-sequencing. Journal of the Science of Food & Agriculture 2018.
3) In this manuscript is not reported which is the actual metabolite production of longan EC or the metabolite production under different lights, this cannot be deduced by the up or -downregulation of metabolic pathways or gene expression, the reviewer suggests to add a supplementary table with the major metabolites produced by longan EC in control condition, it would be a great improvment if a metabolic comparison of the longan Ec under different lights could be reported.Reply: Reference [23] has published polysaccharide, biotin, alkaloid, and flavonoid contents of longan ECs in control, blue and white light. Therefore, this manuscript can cite only these data. Carotenoid contents have been determined in the current manuscript. Metabolism contents of longan ECs under different qualities are shown in Table 1.
[23] Li, H.; Chen, X.; Wang, Y.; Yao, D.; Lin, Y.; Lai, Z., Exploration of the effect of blue light on microRNAs involved in the accumulation of functional metabolites of longan embryonic calli through RNA-sequencing. Journal of the Science of Food & Agriculture 2018.
Table 1 Metabolism contents of longan ECs under different qualities of light
Light quality | Light intensity (µmol·m-2·s-1) | Photoperiod (h) | Polysaccharide contents (mg·g -1 DW) | Biotin contents (mg·g-1 DW) | Alkaloid contents (mg·g-1 DW) | Flavonoid contents (mg·g-1 DW) | Carotenoid contents (mg·g-1 DW) |
Dark | 0 | 47.130a | 0.211a | 15.84a | 6.91a | 11.32a | |
Blue | 32 | 12 | 53.573b | 3.578c | 21.23c | 9.18c | 17.82c |
White | 32 | 12 | 47.662a | 2.038b | 20.46b | 7.91b | 12.65b |
4) Antioxidant enzymes are high in EC under blue light, this finding is interesting, but we do not know whether stress symptoms occurred or the antioxidant machinery is enought efficient to counteract the stress, do not you think that to perform an MDA assay or an ROS detection, might be a step forward?
Reply: We have determined the MDA contents.
The activities of SOD and POD and the concentration of H2O2 were the highest in the blue light treatment group, followed by the white light and dark treatment groups, indicating that the light activated the longan EC enzyme antioxidant system (Figure 1B-1D and Table S2-S4). In addition, we found no significant differences in the MDA contents under dark, blue and white light. Therefore, these results indicated that the antioxident machinery in longan ECs had a major role in light conditions (Figure 1E and Table S5). (manuscript lines 105-110)
Minor text editing is required
Line 30. Delete the word “specialty”
Line 33 change word…. Physiological to physiologically
Line 34. Insert “to humans” after “benefits”
Line 38. Remove repeated word “tyrosine” or replace it with appropriate word
Line 43. Replace “functional metabolites” synthesis and accumulation of functional metabolites
Line 45. Replace “plant functional metabolites” to “production of plant functional metabolites”
Line 62. Change “Tea tree” to tea plant
Line 64. “explain how light signals affect plant morphogenesis and functional metabolites” change to “explore the affect of light on plant morphogenesis and production of functional metabolites”
Line 65. Change “retranscription” to “transcription” also “,most” to “many”
Line 66. Change “in light and” to “of genes as compared to”
Line 69. Change “that light signals affect” to “the effect of light on”
Line 119-120. 65, 116, and 948 to 66, 451 and 578 reads, respectively. Is this read number in Millions? Kindly check it with original data and mention the units.
Line 222. Make italics, via to via
Author Response
Responses to the reviewers’ comments
The reviewer’s questions have been provided below, followed by our point-by-point responses. All of the modifications are highlighted in yellow in the revised manuscript. We appreciate the reviewers’ and editor’s insightful comments and are grateful for the time they have spent reviewing our manuscript.
Reviewer #2
Line 30. Delete the word “specialty”
Line 33 change word…. Physiological to physiologically
Line 34. Insert “to humans” after “benefits”
Line 38. Remove repeated word “tyrosine” or replace it with appropriate word
Line 43. Replace “functional metabolites” synthesis and accumulation of functional metabolites
Line 45. Replace “plant functional metabolites” to “production of plant functional metabolites”
Line 62. Change “Tea tree” to tea plant
Line 64. “explain how light signals affect plant morphogenesis and functional metabolites” change to “explore the affect of light on plant morphogenesis and production of functional metabolites”
Line 65. Change “retranscription” to “transcription” also “most” to “many”
Line 66. Change “in light and” to “of genes compared to”
Line 69. Change “that light signals affect” to “the effect of light on”
Line 119-120. 65, 116, and 948 to 66, 451 and 578 reads, respectively. Is this read number in Millions? Kindly check it with original data and mention the units.
Line 222. Make italics, via to via
Reply: We have corrected the errors in these sentences.
English language has been rewritten and we hope they may have a higher quality than before. We thank American Journal Experts for English language editing.